SciPost Physics

# Physical and unphysical regimes
# of self-consistent many-body perturbation theory

K. Van Houcke[1], E. Kozik[2], R. Rossi[1,3,4,5], Y. Deng[6,7], and F. Werner[8*]

**1** Laboratoire de Physique de l'École normale supérieure, ENS - Université PSL, CNRS,
Sorbonne Université, Université Paris Cité, 75005 Paris, France
**2** Physics Department, King's College, London WC2R 2LS, United Kingdom
**3** Center for Computational Quantum Physics, Flatiron Institute, New York, NY 10010, USA
**4** CNRS, LPTMC, Sorbonne Université, 75005 Paris, France
**5** Institute of Physics, EPFL, 1015 Lausanne, Switzerland
**6** National Laboratory for Physical Sciences at Microscale and Department of Modern Physics,
University of Science and Technology of China, Hefei, Anhui 230026, China
**7** Shanghai Research Center for Quantum Science, Shanghai 201315, China
**8** Laboratoire Kastler Brossel, ENS - Université PSL, CNRS, Collège de France, Sorbonne Université,
75005 Paris, France
* werner@lkb.ens.fr

April 11, 2024

## Abstract

In the standard framework of self-consistent many-body perturbation theory, the skeleton series for the self-energy is truncated at a finite order $N$ and plugged into the Dyson equation, which is then solved for the propagator $G_N$. We consider two examples of fermionic models, the Hubbard atom at half filling and its zero space-time dimensional simplified version. First, we show that $G_N$ converges when $N \to \infty$ to a limit $G_\infty$, which coincides with the exact physical propagator $G_{\text{exact}}$ at small enough coupling, while $G_\infty \neq G_{\text{exact}}$ at strong coupling. This follows from the findings of [1] and an additional subtle mathematical mechanism elucidated here. Second, we demonstrate that it is possible to discriminate between the $G_\infty = G_{\text{exact}}$ and $G_\infty \neq G_{\text{exact}}$ regimes thanks to a criterion which does not require the knowledge of $G_{\text{exact}}$, as proposed in [2].

# 1 Introduction

Self-consistent perturbation theory is a particularly elegant and powerful approach in quantum many-body physics [3–5]. The single-particle propagator $G$ is expressed through the Dyson equation

$$G^{-1} = G_0^{-1} - \Sigma \tag{1}$$

in terms of the non-interacting propagator $G_0$ and the self-energy $\Sigma$, which itself is formally expressed in terms of $G$ through the skeleton series,

$$\Sigma = \Sigma_{\text{bold}}[G] \equiv \sum_{n=1}^{\infty} \Sigma_{\text{bold}}^{(n)}[G] \tag{2}$$

where $\Sigma_{\text{bold}}^{(n)}[G]$ is the sum of all skeleton self-energy Feynman diagrams of order $n$ (these diagrams are built with bold propagator lines representing $G$, and remain connected if one cuts one or two $G$-lines).

The standard procedure for solving Eqs. (1,2) is to truncate the skeleton series at a finite order $N$, and to look for the solution $G_N$ of the self-consistency equation[1]

$$G_N^{-1} = G_0^{-1} - \Sigma_{\text{bold}}^{(\leq N)}[G_N] \tag{3}$$

with

$$\Sigma_{\text{bold}}^{(\leq N)} := \sum_{n=1}^{N} \Sigma_{\text{bold}}^{(n)}.$$

The natural expectation is that one obtains the exact propagator by sending the truncation order to infinity: $G_N \to G_{\text{exact}}$ for $N \to \infty$.

However, as was discovered in [1], the series $\Sigma_{\text{bold}}^{(\leq N)}[G_{\text{exact}}]$ can converge when $N \to \infty$ to a result which differs from the exact physical self-energy $\Sigma_{\text{exact}} = G_0^{-1} - G_{\text{exact}}^{-1}$. This misleading convergence phenomenon was observed for three fermionic textbook models —Hubbard atom, Anderson impurity model, and half-filled 2D Hubbard model— in a region of the parameter space (at and around half filling, at strong interaction and low temperature). $G_{\text{exact}}$ was computed with a numerically exact quantum Monte Carlo method, and the skeleton series was evaluated up to $N = 6$ or 8 by diagrammatic Monte Carlo [6]. Numerous works [2,7–15] have studied various aspects of the problem found in [1], as well as the related divergences of irreducible vertices ([9,11,12,14,16–20] and Refs. therein). In particular, Ref. [8] introduced an exactly solvable toy model, which has the structure of a fermionic model in zero space-time dimensions, and features the misleading convergence problem of [1], as well as the related multivaluedness of the Luttinger-Ward functional also discovered in [1].

In this article, we study the consequences of this problem for the sequence $G_N$, which is the crucial question in the most relevant cases where $G_{\text{exact}}$ is unknown. For the toy model of [8], we find that $G_N$ converges when $N \to \infty$ to a limit $G_\infty$ which differs from $G_{\text{exact}}$ at strong coupling; for the Hubbard atom, our numerical data strongly indicate that such misleading convergence of the sequence $G_N$ also occurs at large coupling and half filling. This misleading convergence of $G_N$ is the first result reported in this article. Secondly, we present data, again for the toy model of [8] and for the Hubbard atom, demonstrating that a criterion proposed in [2] enables one to discriminate between the $G_\infty \neq G_{\text{exact}}$ and $G_\infty = G_{\text{exact}}$ regimes without using the knowledge of $G_{\text{exact}}$.

---

[1]We assume that the solution $G_N$ of (3) is unique, or at least that there is no difficulty in identifying a unique potentially physical solution (*e.g.*, by starting from the weakly interacting limit where $G_N \to G_0$, and following the solution as a function of interaction strength).

The misleading convergence of $G_N$ reported here is a non-trivial fact. It comes from a subtle mathematical mechanism (as we will see), and does not merely follow from the misleading convergence of $\Sigma_{\text{bold}}^{(\leq N)}[G_{\text{exact}}]$ discovered in [1]. Indeed, a naive reasoning would suggest that if the misleading convergence of $\Sigma_{\text{bold}}^{(\leq N)}[G_{\text{exact}}]$ takes place, then $G_N$ should not converge at all.[2]

We restrict here to the scheme (1,2) where $G$ is the only bold element (as in, *e.g.*, Ref. [21]). Nevertheless, our findings may also be relevant to other schemes containing additional bold elements, such as a bold interaction line $W$, or a bold pair propagator line $\Gamma$. The scheme built with $G$ and $W$ is natural for Coulomb interactions, and is widely used for solids and molecules with a truncation order $N = 1$ (the $GW$ approximation) and sometimes with $N = 2$ (see, *e.g.*, Refs. [22–25]), while for several paradigmatic lattice models, bold diagrammatic Monte Carlo (BDMC) made it possible to reach larger $N$ and claim a small residual truncation error [26–29]. The scheme built with $G$ and $\Gamma$ is natural for contact interactions; truncation at order $N = 1$ then corresponds to the self-consistent T-matrix approximation [30–32], and precise large-$N$ results were obtained by BDMC in the normal phase of the Hubbard model [33, 34] and of the unitary Fermi gas [35–37]. Other BDMC results were obtained for models of coupled electrons and phonons, where it is natural to introduce a bold phonon propagator [26, 38], and for frustrated spins [39–41]. Schemes containing three- or four-point bold vertices were also employed, to construct extensions of dynamical mean-field theory [18, 42].

## 2 Zero space-time dimensional toy-model

### 2.1 Definitions and reminders

We begin with some reminders from [8] (see [8] for the derivations). While fermionic many-body problems can be represented by a functional integral over Grassmann *fields*, which depend on $d$ space coordinates and one imaginary time coordinate [4, 43], in this simplified toy model the Grassmann fields are replaced with Grassmann *numbers* $\varphi_s$ and $\bar{\varphi}_s$ that do not depend on anything, apart from a spin index $s \in \{\uparrow, \downarrow\}$. The partition function, the action and the propagator are then defined by

$$Z = \int \left( \prod_s d\varphi_s \, d\bar{\varphi}_s \right) e^{-S}$$

$$S = -\mu \sum_s \bar{\varphi}_s \varphi_s + U \, \bar{\varphi}_\uparrow \varphi_\uparrow \bar{\varphi}_\downarrow \varphi_\downarrow$$

$$G = -\frac{1}{Z} \int \left( \prod_s d\varphi_s \, d\bar{\varphi}_s \right) \varphi_{s'} \bar{\varphi}_{s'} \, e^{-S},$$

the dimensionless parameters $\mu$ and $U$ being the analogs of chemical potential and interaction strength. Since $G$ is spin-independent, we omit its spin index. We restrict for convenience to $\mu > 0$ (changing the sign of $\mu$ essentially amounts to the change of variables $\varphi \leftrightarrow \bar{\varphi}$) and to $U < 0$ (as in [8]).

The coefficients of the skeleton series have the analytical expression

$$\Sigma_{\text{bold}}[G] = \sum_{n=1}^{\infty} a_n G^{2n-1} U^n \quad \text{with} \quad a_n = \frac{(-1)^{n-1}(2n-2)!}{n!(n-1)!}.$$

---

[2]The naive reasoning goes as follows: If $G_N$ would converge to some $G_\infty$ for $N \to \infty$, then, from (3), one can expect $G_\infty^{-1} = G_0^{-1} - \Sigma_{\text{bold}}[G_\infty]$, and hence, assuming unicity of the solution of the Dyson equation, $G_\infty = G_{\text{exact}}$. Thus $\Sigma_{\text{bold}}[G_{\text{exact}}] = \Sigma_{\text{exact}}$, in contradiction with the misleading convergence of $\Sigma_{\text{bold}}^{(\leq N)}[G_{\text{exact}}]$.

It is convenient to work with rescaled variables, multiplying propagators with $\sqrt{|U|}$ and dividing self-energies with the same factor,

$$g := G\sqrt{|U|}, \qquad \sigma := \Sigma/\sqrt{|U|}. \tag{4}$$

The rescaled skeleton series is then given by

$$\sigma_{\text{bold}}(g) = \sum_{n=1}^{\infty} \sigma_{\text{bold}}^{(n)}(g) \quad \text{with} \quad \sigma_{\text{bold}}^{(n)}(g) = a_n(-1)^n g^{2n-1}$$

and accordingly $\sigma_{\text{bold}}^{(\leq N)}(g) \equiv \sum_{n=1}^{N} \sigma_{\text{bold}}^{(n)}(g)$.

The exact self-energy and propagator are given by

$$\begin{aligned}
\sigma_{\text{exact}}(g_0) &= -g_0 \\
g_{\text{exact}}(g_0) &= \frac{g_0}{1+g_0^2}
\end{aligned}$$

in terms of the rescaled free propagator $g_0 := \sqrt{|U|}\,G_0 = \sqrt{|U|}/\mu$.

If one evaluates the skeleton series at the exact $G$, one obtains the correct physical self-energy for $|U| < \mu^2$ and an incorrect result for $|U| > \mu^2$. This is directly related to the fact that the self-energy functional (which reduces to a function in this toy model) has two branches,

$$\sigma^{(\pm)}(g) = \frac{-1 \pm \sqrt{1-4g^2}}{2g} \tag{5}$$

as represented in Fig. 1 (this corresponds to the derivative of the two branches of the Luttinger-Ward functional, restricting to spin-independent $G$ for simplicity). The physical branch is the $(+)$ branch for $g_0 < 1$, and the $(-)$ branch for $g_0 > 1$; *i.e.*, $\sigma_{\text{exact}}(g_0) = \sigma^{(\text{sign}(1-g_0))}(g_{\text{exact}}(g_0))$. On the other hand, the skeleton series, evaluated at the exact physical propagator, always converges to the $(+)$ branch; *i.e.*, $\sigma_{\text{bold}}(g_{\text{exact}}(g_0)) = \sigma^{(+)}(g_{\text{exact}}(g_0))$ for all $g_0 > 0$.

Note that $\sigma_{\text{bold}}(g)$ is the expansion of $\sigma^{(+)}(g)$ in powers of $g$, and thus from (5) the convergence radius of the series $\sigma_{\text{bold}}(g)$ is $1/2$.

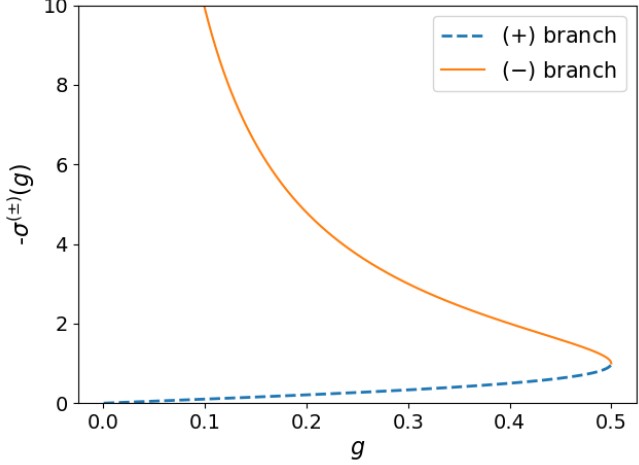

Figure 1: The two branches of the self-energy as a function of the full propagator, for the toy model in zero space-time dimensions. The skeleton series converges up to $g = 1/2$ and coincides with the $(+)$ branch: $\sigma_{\text{bold}}(g) = \sigma^{(+)}(g)$ for $g \leq 1/2$.

## 2.2 Limit of the skeleton sequence

We now go beyond Ref. [8] and study the "skeleton sequence" $G_N$ defined by Eq. (3). Rescaling variables as in (4), in particular setting $g_N := G_N \sqrt{|U|}$, Equation (3) becomes

$$\frac{1}{g_N} = \frac{1}{g_0} - \sigma_{\text{bold}}^{(\leq N)}(g_N). \tag{6}$$

This equation is readily solved for $g_N$ numerically: The solutions are roots of a polynomial of order $2N$, and we observe that there is a unique real positive root, which we take to be $g_N$ (recall that the exact $g$ is always real and positive); alternatively, we solved Eq. (6) by iterations (with a damping procedure described in the next section), and we found convergence to this same $g_N$. We find that

- for $g_0 < 1$, $g_N \underset{N \to \infty}{\longrightarrow} g_{\text{exact}}(g_0)$

- for $g_0 > 1$, $g_N \underset{N \to \infty}{\longrightarrow} g_\infty \neq g_{\text{exact}}(g_0)$

*i.e.*, the skeleton sequence converges to the correct physical result below a critical coupling strength, and to an unphysical result above it.

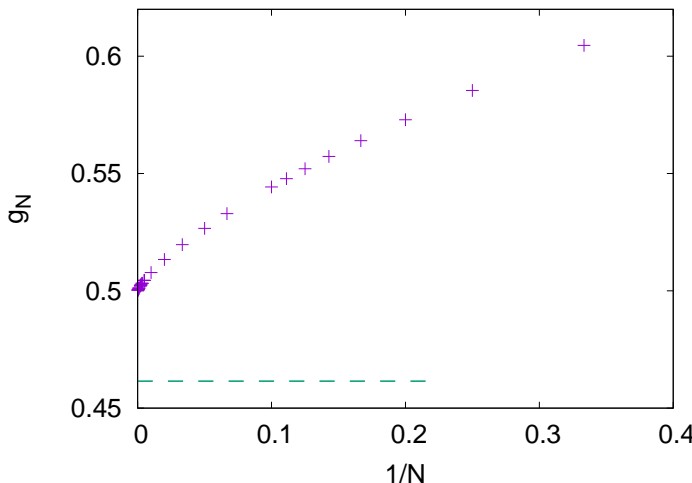

Figure 2: *Illustrative example of misleading convergence of the skeleton sequence for the toy model.* The rescaled propagator $g_N$, obtained from the self-consistency equation with the skeleton series truncated at order $N$, converges for $N \to \infty$ to the limit 0.5, which differs from the exact result (dashed line). This happens when the rescaled free propagator $g_0 > 1$ (here, $g_0 = 1.5$).

Let us focus on the regime $g_0 > 1$, where the convergence to an unphysical result takes place (as demonstrated in Fig. 2). The fact that the skeleton sequence converges at all in this regime is non-trivial. The value of the unphysical limit $g_\infty = 1/2$ of the skeleton sequence $g_N$ is equal to the radius of convergence of the skeleton series $\sigma_{\text{bold}}(g)$. This is not a coincidence, and the reason for this self-tuning towards the convergence radius becomes clear from Fig. 3: For a large truncation order, the curve representing the truncated skeleton series as a function of $g$ becomes an almost vertical line above the position of the convergence radius ($g = 1/2$), so that it intersects the Dyson-equation curve near this value of $g$. It also becomes clear that we are in an unusual situation where

$$\lim_{N \to \infty} \sigma_{\text{bold}}^{(\leq N)}(g_N) \neq \lim_{N \to \infty} \sigma_{\text{bold}}^{(\leq N)}(g_\infty) \equiv \sigma_{\text{bold}}(g_\infty). \tag{7}$$

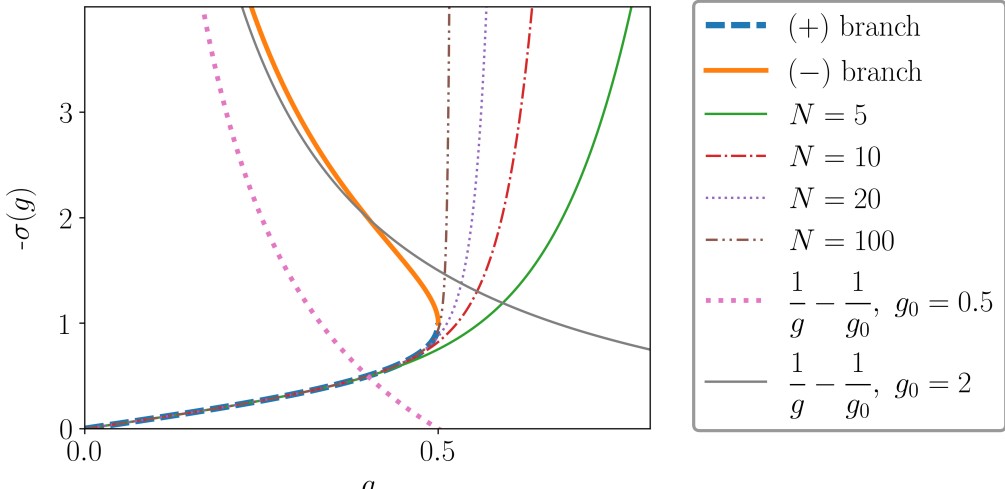

Figure 3: *Explanation for the misleading convergence.* The two branches of the self-energy $\sigma^{(\pm)}(g)$, together with the partial sums of the skeleton series $\sigma_{\text{bold}}^{(\leq N)}(g)$ for different values of the truncation order $N$. Also shown is the curve corresponding to the Dyson equation, $-\sigma = 1/g - 1/g_0$. This Dyson-equation curve intersects $\sigma_{\text{bold}}^{(\leq N)}(g)$ at $g = g_N$, whereas the exact propagator $g = g_{\text{exact}}$ is given by the intersection of the Dyson-equation curve with the physical branch $\sigma^{(\text{sign}(1-g_0))}(g)$. It appears clearly that for $g_0 < 1$, $g_N$ converges to the exact $g$, while for $g_0 > 1$, $g_N$ always tends to $1/2$, the convergence radius of the skeleton series.

## 2.3 Diagnosing the misleading convergence

In the general case where $G_{\text{exact}}$ is unknown, when one observes numerically that $G_N$ converges to some limit, one needs a way to tell whether this limit is equal to $G_{\text{exact}}$, *i.e.*, whether the result can be trusted. To this end, we consider

$$\Sigma_{N,\xi} := \sum_{n=1}^{N} \Sigma_{\text{bold}}^{(n)}[G_N] \; \xi^n. \tag{8}$$

Assuming that $G_N \to G_\infty$ for $N \to \infty$, the following criterion [2] is a sufficient condition for $G_\infty$ to be equal to $G_{\text{exact}}$:

$$\left\{ \begin{array}{l} \textit{There exists } \epsilon > 0 \textit{ such that:} \\ \textit{For any } \xi \textit{ in the disc } \mathcal{D} = \{\, |\xi| < 1 + \epsilon \,\}, \; \Sigma_{N,\xi} \textit{ converges for } N \to \infty; \\ \textit{moreover, this sequence is uniformly bounded for } \xi \in \mathcal{D}. \end{array} \right\} \tag{9}$$

The derivation of this criterion is contained in [2], and its main steps are reproduced in the Appendix for convenience.

For all practical purpose, we expect the criterion (9) to be essentially equivalent to the following simpler one:

$$\textit{There exists } \xi > 1 \textit{ such that } \; \Sigma_{N,\xi} \textit{ converges for } N \to \infty. \tag{10}$$

Indeed, (9) implies (10), and a situation where (10) would hold while (9) would not hold seems unlikely to occur. In what follows we will use the simplified criterion (10). We also introduce an extra factor $1/\xi^{N_0}$ in the definition (8) of $\Sigma_{N,\xi}$, where the value of $N_0$ will be conveniently chosen; such an $N$-independent factor does not matter for the criterion (it does not change whether or not the sequence $\Sigma_{N,\xi}$ converges).

For the toy-model, this means that assuming $g_N \to g_\infty$ for $N \to \infty$, a sufficient condition for $g_\infty$ to be equal to the correct physical $g_{\text{exact}}(g_0)$ is that there exists $\xi > 1$ such that

$$\sigma_{N,\xi} := \sum_{n=1}^{N} \sigma_{\text{bold}}^{(n)}(g_N) \, \xi^{n-1} = \sigma_{\text{bold}}^{(\leq N)}(g_N \sqrt{\xi}) / \sqrt{\xi}$$

converges for $N \to \infty$. As illustrated in Fig. 4, this criterion indeed enables one to detect the misleading convergence for $g_0 > 1$, and to trust the result for $g_0 < 1$.[3]

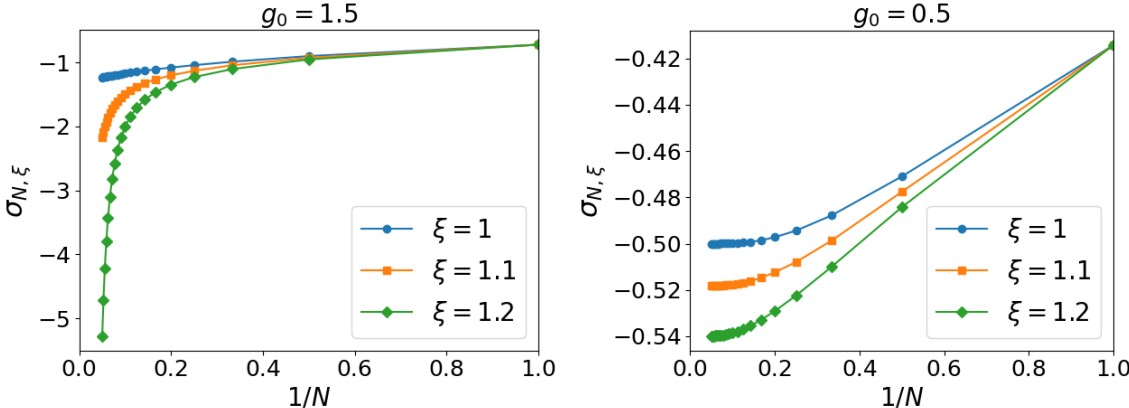

Figure 4: *Detecting the misleading convergence for the toy model.* Introducing a finite $\xi$, the sequence becomes divergent which enables one to detect the problem (left panel), or remains convergent which enables one to trust the result (right panel).

## 3  Hubbard atom

We turn to the single-site Hubbard model, defined by the grand-canonical Hamiltonian $-\mu \sum_s n_s + U \, n_\uparrow n_\downarrow$. The propagator can be expressed as a functional integral over $\beta$-antiperiodic Grassmann fields [4, 43],

$$G_s(\tau) = -\langle \varphi_s(\tau) \bar{\varphi}_s(0) \rangle_S \equiv -\frac{\int \mathcal{D}\varphi \, \mathcal{D}\bar{\varphi} \; \varphi_s(\tau) \bar{\varphi}_s(0) \, e^{-S}}{\int \mathcal{D}\varphi \, \mathcal{D}\bar{\varphi} \; e^{-S}} \tag{11}$$

with the action

$$S = \int_0^\beta d\tau \left[ -\sum_s \bar{\varphi}_s(\tau)(G_0^{-1} \varphi_s)(\tau) + U \, (\bar{\varphi}_\uparrow \bar{\varphi}_\downarrow \varphi_\downarrow \varphi_\uparrow)(\tau) \right] \tag{12}$$

and

$$G_0^{-1} = \mu - \frac{d}{d\tau}. \tag{13}$$

We restrict for simplicity to the half-filled case $\mu = U/2$, which should be the most dangerous case, since it is at and around half-filling that the misleading convergence of $\Sigma_{\text{bold}}[G_{\text{exact}}]$ was discovered in [1]. We use the BDMC method [6, 35, 44, 45] to sum all skeleton diagrams

---

[3]For the toy model, the criterion is easily understood from Fig. 3. For $g_0 > 1$, $g_N \to 1/2$ for $N \to \infty$, so that for any $\xi > 1$, $\lim_{N \to \infty} g_N \sqrt{\xi}$ is strictly larger than the convergence radius $1/2$, leading to the divergence of $\sigma_{\text{bold}}^{(\leq N)}(g_N \sqrt{\xi})$. On the contrary, for $g_0 < 1$, the $g_N$'s stay at a finite distance on the left of the convergence radius $1/2$.

and solve the self-consistency equation (3) for truncation orders $N \leq 8$ (note that at half filling, $\Sigma_{\text{bold}}^{(n)} = 0$ for all odd $n > 1$).

The first question is whether the skeleton sequence $G_N$ can also converge to an unphysical result, or equivalently, whether $\Sigma_{\text{bold}}^{(\leq N)}[G_N] =: \Sigma_N$ can converge to an unphysical result. Let us first consider the double occupancy

$$D = \langle n_\uparrow n_\downarrow \rangle = U^{-1} \operatorname{tr}(\Sigma G)$$

and the corresponding sequence $D_N := U^{-1} \operatorname{tr}(\Sigma_N G_N)$. At large enough $U$, our data strongly indicate that this sequence does converge (albeit slowly) towards an unphysical result, see left panel of Fig. 5. For small enough $U$, there is a fast convergence to the correct result, see right panel of Fig. 5.

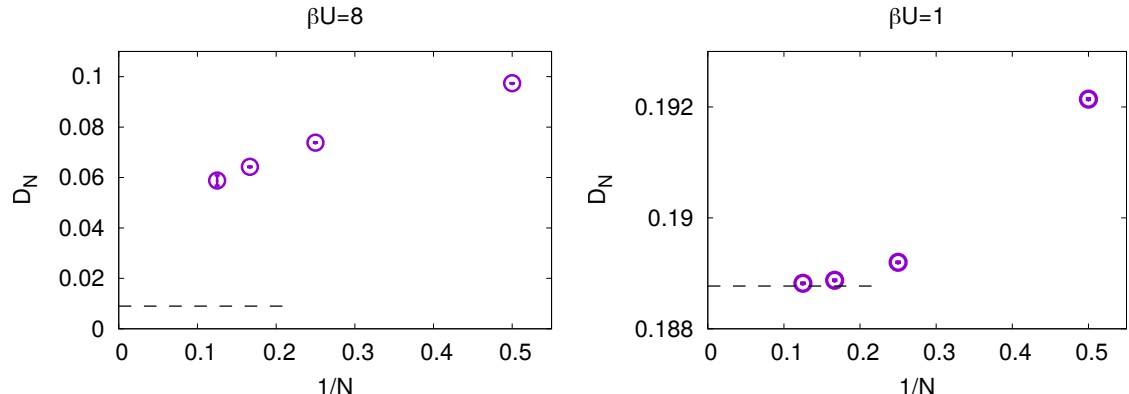

Figure 5: For the Hubbard atom at half filling, the double occupancy, as obtained from the skeleton sequence, converges to an unphysical result for large $U$ (left panel) and to the correct result for small enough $U$ (right panel) when the truncation order $N \to \infty$ (dashed line: exact result).

The next question is whether the criterion (10) enables us to discriminate between these two situations. We therefore plot the sequence $\Sigma_{N,\xi}$ in Figs. 6 and 7. We only show the imaginary part because in the considered half-filled case, the real part of $\Sigma_N(\omega_n)$ automatically equals $U/2$; moreover we focus for simplicity on the lowest Matsubara frequency $\omega_0 = \pi/\beta$, and we choose $N_0 = 2$.

For $\xi = 1$, $\Sigma_{N,\xi}$ reduces to the original skeleton sequence $\Sigma_N$, and the behavior is similar to the double occupancy: The sequence appears to converge, albeit slowly, towards an unphysical result for $\beta U = 8$ (Fig. 6), while fast convergence to the correct physical result takes place for $\beta U = 1$ (Fig. 7). For $\xi > 1$, the sequence does not appear to converge any more for $\beta U = 8$, see Fig. 6: The data do not satisfy the criterion, indicating that the results cannot be trusted in this case. In contrast, for $\beta U = 1$, the criterion enables one to validate the results, since the sequence remains convergent at $\xi > 1$, see Fig. 7.

Regarding the choice of $\xi$, it should be neither too small in order to have an effect at the accessible orders, nor too large to avoid making the criterion too conservative. More precisely, $\xi - 1$ should not be too small, so that $\xi^N$ differs significantly from 1 (and hence $\Sigma_{N,\xi}$ differ significantly from $\Sigma_{N,1} = \Sigma_N$) at the largest accessible order $N_{\text{max}}$. This necessity to work with a finite $\xi - 1$ implies that the criterion is conservative: It leads to discarding results in a region of the parameter space near but outside the misleading-convergence region.[4] For $N_{\text{max}} = 8$, the choices $\xi - 1 = 0.1$ and $0.2$ are *a priori* large enough (since $\xi^8 \approx 2$ and $4$) and Fig. 6 confirms

---

[4]The required $\xi - 1$ scales as $1/N_{\text{max}}$; for the toy model this leads to discarding results not only in the misleading-convergence region $g_0 > 1$, but also for $0 \leq 1 - g_0 \lesssim 1/N_{\text{max}}$.

that they allow to detect the divergence of $\Sigma_{N,\xi}$ in the misleading-convergence regime, while on the other hand Fig. 7 shows that $\beta U = 1$ is at a sufficient distance from the misleading-convergence region for $\Sigma_{N,\xi}$ to remain convergent for these $\xi$ values.

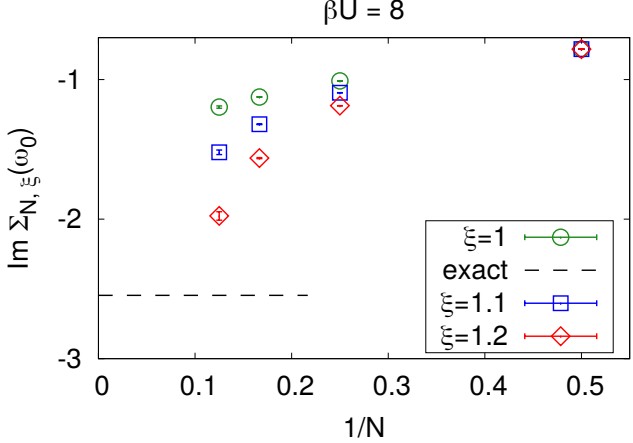

Figure 6: For the half-filled Hubbard atom at large coupling, the original skeleton sequence ($\xi = 1$) converges to an unphysical result. At $\xi > 1$, the sequence does not converge any more: The data do not satisfy the criterion.

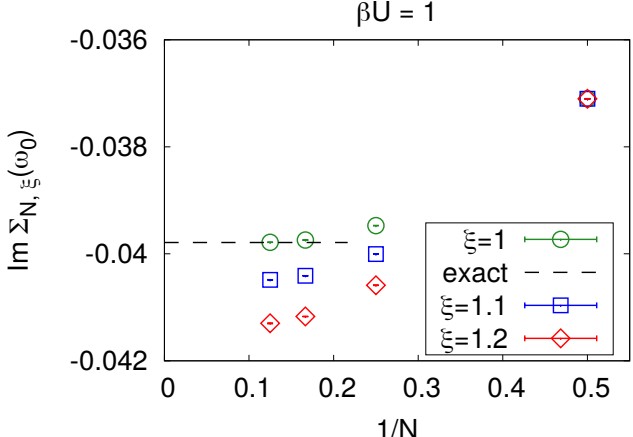

Figure 7: For the half-filled Hubbard atom at small enough coupling, the original skeleton sequence ($\xi = 1$) converges to the correct physical result. At $\xi > 1$, the sequence remains convergent: The criterion enables one to trust the result.

We remark that when solving Eq. (3) by iterations, in the case where the convergence to the unphysical result for $N \to \infty$ occurs, convergence as a function of iterations at fixed $N$ only takes place if we use a damping procedure, where $G_N$ at iteration $(i + 1)$ is obtained as $G_N^{(i+1)} = [G_0^{-1} - \Sigma^{(i)}]^{-1}$ with $\Sigma^{(i)}$ a weighted average of $\Sigma_{\text{bold}}^{(\leq N)}[G_N^{(i)}]$ and $\Sigma^{(i-1)}$, while the fixed point is unstable for the undamped iterative procedure $\Sigma^{(i)} := \Sigma_{\text{bold}}^{(\leq N)}[G_N^{(i)}]$. Such a damping procedure is commonly used in BDMC where it also reduces the statistical error [45, 46]. In the toy model, one can easily show that an increasingly strong damping is required when $N$ is increased, because for $N \to \infty$, the slope $[d\sigma_{\text{bold}}^{(\leq N)}(g)/dg]_{g=g_N}$ diverges, making the undamped iterative procedure unstable. This observation could also be useful for misleading-convergence detection.

Finally, we comment on the link with the multivaluedness of the self-energy functional $\Sigma[G]$ (*i.e.*, of the Luttinger-Ward functional). In [1], the misleading convergence of the skeleton series was found to be towards an unphysical branch of the self-energy functional, in the sense that if Eqs. (11,12) are viewed as a mapping $G_0 \mapsto G[G_0]$, then there exists $G_{0,\text{unphys}}$ such that $\Sigma_{\text{bold}}[G_{\text{exact}}] = G_{0,\text{unphys}}^{-1} - G_{\text{exact}}^{-1}$ and $G[G_{0,\text{unphys}}] = G_{\text{exact}} \equiv G[G_0]$. As noted in [1], this $G_{0,\text{unphys}}$ does not belong to the set of physical bare propagators which are of the form (13) for some value of chemical potential; therefore, by looking at $G_{0,\text{unphys}}$, one can tell that the result is on an unphysical branch, and hence detect the misleading convergence of the skeleton series. In contrast, the misleading convergence of the sequence $G_N$ reported here cannot be detected in this way. Indeed, the self-consistency equation (3) is enforced with the original physical $G_0$.

## 4 Conclusion

We have demonstrated that there is a regime where the solution of self-consistent many-body perturbation theory converges to an unphysical result in the limit of infinite truncation order of the skeleton series. This surprising breakdown of the standard framework results from the findings of [1] combined with an additional subtle mathematical mechanism which we have elucidated by analyzing the zero space-time dimensional model of [8]. In this problematic regime, lowest order calculations can be off by one order of magnitude, but access to higher orders enables one to detect the problem numerically through the divergence of a slightly modified sequence, whereas seeing convergence of this modified sequence enables one to rule out misleading convergence and to trust the result, as proposed in [2] and demonstrated here for the Hubbard atom. Such a proof of principle is relevant for many-body problems in regimes where, in spite of important progress with non self-consistent frameworks [47–68] (for which it was shown that misleading convergence generically does not occur [2]) and with strong-coupling expansions [69–71], self-consistent BDMC remains among the state of the art approaches. In particular, during finalization of the present manuscript, its main findings have been used as a basis to discriminate between physical and unphysical BDMC results for the doped two-dimensional Hubbard model at strong coupling in a non-Fermi liquid regime [72].

## Acknowledgments

We thank N. Prokof'ev, B. Svistunov and L. Reining for useful discussions and comments.

**Funding information**  We acknowledge support from ERC (*Critisup2*, H2020 Adv-743159) (F.W.), ANR (*LODIS*, ANR-21-CE30-0033) (K.V.H. and F.W.), the Simons Foundation through the Simons Collaboration on the Many Electron Problem and EPSRC (EP/P003052/1) (E.K.), the National Natural Science Foundation of China (grant No. 11625522) and the Science and Technology Committee of Shanghai (grant No. 20DZ2210100) (Y.D.). The Flatiron Institute is a division of the Simons Foundation.

## Appendix: Main steps of the derivation of the criterion

For convenience, we reproduce here the main steps of the derivation of the criterion (9), see [2,73] for more details. For definiteness, we consider the Hubbard model at finite temperature; the reasoning is also directly applicable to the Hubbard atom (by removing the position variable) and to the zero space-time dimensional toy model (by also removing the imaginary time variable). Let $\Sigma_{\infty,\xi} := \lim_{N\to\infty} \Sigma_{N,\xi}$. By making use of Morera's theorem, Cauchy's integral formula and the dominated convergence theorem, the condition (9) allows one to show the key property

$$\Sigma_{\infty,\xi} = \sum_{n=1}^{\infty} \Sigma_{\text{bold}}^{(n)}[G_\infty]\, \xi^n, \quad \forall \xi \in \mathcal{D}. \tag{14}$$

Setting $\xi = 1$ in (14) yields[5]

$$\lim_{N\to\infty} \Sigma_{\text{bold}}^{(\leq N)}[G_N] = \Sigma_{\text{bold}}[G_\infty]. \tag{15}$$

Substituting (15) into (3) yields

$$G_\infty^{-1} = G_0^{-1} - \Sigma_{\text{bold}}[G_\infty]. \tag{16}$$

The next step is to consider the action

$$\begin{aligned}
S_{\text{bold}}^{(\xi)} := & -\sum_{\mathbf{r},s}\int_0^\beta d\tau \left[ \bar{\varphi}_s \left( G_\infty^{-1} + \sum_{n=1}^{\infty} \Sigma_{\text{bold}}^{(n)}[G_\infty]\, \xi^n \right) \varphi_s \right](\mathbf{r},\tau) \\
& + \xi\, U \sum_{\mathbf{r}}\int_0^\beta d\tau\, (\bar{\varphi}_\uparrow \bar{\varphi}_\downarrow \varphi_\downarrow \varphi_\uparrow)(\mathbf{r},\tau)
\end{aligned}$$

and the corresponding propagator $G_{\text{bold}}^{(\xi)}(\mathbf{r},\tau) := -\langle \varphi_s(\mathbf{r},\tau)\, \bar{\varphi}_s(\mathbf{0},0)\rangle_{S_{\text{bold}}^{(\xi)}}$. The action $S_{\text{bold}}^{(\xi)}$ is designed in such a way that

$$\left.\frac{\partial^n G_{\text{bold}}^{(\xi)}}{\partial \xi^n}\right|_{\xi=0} = 0, \quad \forall n \geq 1. \tag{17}$$

Obviously, $G_{\text{bold}}^{(\xi=0)} = G_\infty$. On the other hand, (16) implies that $S_{\text{bold}}^{(\xi=1)}$ is equal to the physical action of the Hubbard model, so that $G_{\text{bold}}^{(\xi=1)} = G_{\text{exact}}$. Now, since $S_{\text{bold}}^{(\xi)}$ depends analytically on $\xi$ in $\mathcal{D}$, we expect (at least for fermions on a lattice at finite temperature) that one of the following alternatives holds:

 (i) $G_{\text{bold}}^{(\xi)}$ depends analytically on $\xi$ in $\mathcal{D}$

 (ii) $G_{\text{bold}}^{(\xi)}$ has a non-removable singularity at a point $\xi_c \in \mathcal{D}$ (e.g., a phase transition), and $G_{\text{bold}}^{(\xi)}$ is analytic in the disc $\{|\xi| < |\xi_c|\}$.

In case (ii), the convergence radius of the Taylor series of $G_{\text{bold}}^{(\xi)}$ at the origin would be $|\xi_c|$, in contradiction with (17). Hence (i) holds, and we have

$$G_{\text{exact}} = G_{\text{bold}}^{(\xi=1)} = G_{\text{bold}}^{(\xi=0)} + \sum_{n=1}^{\infty} \frac{1}{n!} \left.\frac{\partial^n G_{\text{bold}}^{(\xi)}}{\partial \xi^n}\right|_{\xi=0} = G_{\text{bold}}^{(\xi=0)} = G_\infty.$$

This concludes the derivation of the equality $G_\infty = G_{\text{exact}}$ under the assumption (9).

---

[5]Equation (15) breaks down for the toy model in the problematic regime ($g_0 > 1$), as pointed out in Eq. (7).

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
