# Peer review of "Physical and unphysical regimes of self-consistent many-body perturbation theory"

_SciPost Physics_

## Round 1 · Referee Report · Anonymous · 2021-3-4

Strengths
1) The paper is very well written.
2) It explains in a clear and pedagogical how to apply in practice the test for determining whether a converged bold-diagrammatic Monte Carlo (BDMC) result is physical or not. Using this test in practice may prove very valuable in the future.
Weaknesses
1) It is not stated clearly in the text what the new contribution is that is presented in this manuscript.
2) the paper lacks a clear discussion of conditions under which the proposed test can be used with confidence, and cases where it might be insufficiently robust or discriminative.
Report
The authors discuss the break-down of bold-diagrammatic series at strong couplings. It has been shown in several works in the last years that bold-diagrammatic series can converge to an unphysical solution, which can lead to a misinterpretation of the results. In PHYSICAL REVIEW B93, 161102(R) (2016), two of the authors prove that a bold series that converges to the correct (physical) solution, satisfies a certain mathematical criterion. This criterion can be used to formulate a practical numerical test to discriminate between physical and unphysical solutions of BDMC. Such a test was used successfully in a recent preprint arXiv:2012.06159. In the present manuscript, the authors apply the numerical test to two simplified models: the 0+0-dimensional "toy model", and the 0+1-dimensional Hubbard atom.
The topic of the paper is very important. The diagrammatic Monte Carlo tools have gained in importance in recent years, so having a well defined test for the quality of obtained solutions is paramount for further development and applications. The manuscript is written clearly and in a pedagogical way, and can be easily followed.
However, the paper seems to lack important new conclusions. The mathematical criterion (Eq.7) was proposed elsewhere, and the numerical test that is showcased in this manuscript, has already been used before. Nevertheless, the paper makes up for the lack of novelty by clarity and conciseness. The precise way the numerical test works might not be as easily comprehended from previous publications, and may easily be overlooked as an essential tool for future works. It is a good idea to have a separate publication focused solely on the application of the proposed numerical test. I suggest that the authors improve the abstract and conclusion so that the importance of this work can be more easily appreciated.
Additionally, I feel that the application of the criterion has not been tested in a sufficiently stringent manner.
For example, in Fig.4 the authors only show two cases - g_0=0.5 which is far on the physical side, and g_0=1.5 which is far on the unphysical side. How does the numerical test hold in the near vicinity of g_0=1? Similarly, in the case of the Hubbard atom, the values of coupling chosen are betaU=8 and betaU=1, both far away from the coupling critical for the transition to the unphysical solution.
At least in the 0+0d toy model case where the numerics is inexpensive, it would make sense to investigate the behavior of the xi>1 series at around g_0=1. Can the authors determine the maximal xi>1 for which the series Eq.7 is convergent (say, xi_max)? How does xi_max depend on g_0? It appears that in practice, if xi_max is very close to 1, it may be difficult to reach high enough orders to see any difference from the original series with xi=1 and properly apply the test. As the authors themselves state:
"Regarding the choice of ξ, it should be neither too small in order to have an effect at the accessible orders, nor too large to avoid making the criterion too conservative; "
- for the test to be useful in practice, one must have at least some idea of what a "too large" xi is. Otherwise, the test might lead to false identification of physical solutions as unphysical.
I invite the authors to elaborate on the choice of xi values, and give some guidelines on setting up and interpreting the test in cases when the results of the test might not be as clear-cut.
Finally, my impression is that, for the paper to be reasonably self-contained, some form of a mathematical proof for the criterion stated in Eq.7 (or the weaker statement in Eq.8) should be given. If the criterion can already be anticipated by looking at Fig.3, this should be explained in the text.
Requested changes
1) improve abstract and conclusions - state more clearly what the results are of the present study, or the aim of the publication
2) test the method in a more difficult regime (g_0~1) where the results of the test might not be as clear-cut.
3) determine the maximal xi for which Eq.7 is convergent, as a function of g_0.
4) Give more elaborate guidelines on using the method and avoiding misinterpretation of the results.
5) at least outline the proof of the criterion Eq.7 or briefly explain the reasoning behind it

---

## Round 1 · Referee Report · Anonymous · 2021-3-22

Strengths
-) Very clear description of the problem of unphysical solutions in many-body perturbation theory
-) Introduction of a criterion to distinguish physical from unphysical solutions without the actual knowledge of the solution itself
-) Pedagogical explanation of the new criterion for two very simple models
Weaknesses
-) Unclear use or misuse of the notion of the Luttinger-Ward functional
-) Missing derivation/explanation of the criterion to detect unphysical solutions in bold self-consistent perturbation theory
-) Insufficient clarification of originality/novelty of the results.
Report
In their paper, the authors discuss the important problem of the convergence of self-consistent perturbation theory in different coupling regimes. This is indeed a very important question as this technique is the core of many diagrammatic quantum field theoretical approaches to treat the many-electron (or, more generally, many-particle) problem, in particular for diagrammatic Monte Carlo methods. In the last years, it has been realized that even if the self-consistency cycle for the calculation of the single-particle self-energy converges, it might converge to an unphysical solution, in particular for strong couplings. The authors demonstrate this effect for two simple models, a zero space-time model and the Hubbard atom. Moreover, they introduce a criterion to detect a convergence to an unphysical solution which does not require the knowledge of the self-energy (or the one-particle Green's function) itself.
The paper is well and clearly written and rather easy to digest also for a non-expert reader. However, it is not completely clear what is really new in this manuscript. I think, the authors should indicate more clearly which part of the paper is more kind of a review of older results and what is really new. In this respect, let me stress that I do not consider it as a problem if the amount of new results is rather low since I consider the manuscript as a useful summary of the state of the art in the field.
Having said this I have one major concern about the correctness regarding one issue, which I have already recognized in some previous papers of the authors (e.g., in Ref. 7): There often seem to be a confusion what the Luttinger Ward of a correlated system is. Let me first state that, in principle, one could use *any* functiondal \Sigma[G], for which \Sigma[G_phys]=\Sigma_\phys to construct a self-consistent perturbation theory. This does not has to originate from the Luttinger-Ward functional of the system. In the paper, the functional for the zero space-time model in Eq.(5) is actually the functional of a non-interacting model with a quenched disorder as was described in Ref.[8]. This is, however, not the "correct" functional for a correlated system. The problem cannot be detected at the level of the one-particle Green's function as in this situation non-interacting, disordered and correlated systems can lead to the same result (For instance, the single-particle Green's function of the half-filled Hubbard atom with Coulomb repulsion U, of a non-interacting two-level system with levels +/- U/2 and a binary disorder with disorder strength W=U are equivalent). In fact, if one calculates the two-particle Green's function G_2 from the action and extract the generalized susceptibility via \chi=G_2 - G_1*G_1, this should be equivalent to the solution of the Bethe-Salpeter equation \chi = GG - GG*\Gamma*\chi, where the irreducible vertex \Gamma is the functional derivative of the self-energy functional \Sigma[G] w.r.t. G, i.e., \Gamma=d\Sigma[G] / dG. As far as I can see, this is not the case for the zero space-time model in the manuscript: The functional \Sigma[G] in Eq.(5) is diagonal in the spin and, hence, \Gamma_up_down and \chi_up_down must be 0. Calculating \chi_up_down directly from the action of the zero space-time model, on the other hand, gives a finite \chi_up_down if I am not mistaken. On the contrary, functional (5) yields the correct two-particle Green's function for a system with a quenched disorder as was shown in Ref.[8]. Let me mention that even this does not proof the correctness of the functional \Sigma[G]---in principal one has to calculate all higher-order Green's functions.
The authors should, hence, either clearly state that the functional that they use is that of a binary mixture (which has been discussed extensively in Ref.[8]) in spite of the correlated nature of the system (which will not provide correct results at the two-particle level), or they should modify their action to the action of a zero space-time model with a quenched disorder. Since the latter has been analysed in detail in Ref.[8] the authors should mention this more clearly. Let me stress that I find this point very important since quite some confusion has been produced regarding the "correct" functional \Sigma[G] for correlated systems before.
In this respect, it would be also interesting, if the authors could comment on the relation of their work to the self-consistent scheme for the two branches of \Sigma proposed in Eqs.(26). Moreover, in PRB 98, 235107 (2018), an approximate functional for the correlated Hubbard atom has been proposed in Eqs.(44) and (45) which is based on iterated perturbation theory. Can this be used to get a more realistic approximation for the correlated zero space-time model of the authors than the \Sigma[G] of a disordered system in Eq.(5)?
My second main point is probably easier to address: I think, for completeness, it would be very nice to proved a proof or, at least, a short explanation to the criterion in Eq.(7).
Apart from these main points I found I few minor mistakes and typos:
-) Below Eq.(2): Isn't the self-energy the sum of all *one*-particle irreducible diagrams?
-) Conclusions: In the third line, I think it should be "zero space-time" instead of "zero space-space".
Requested changes
1) Discuss the validity of the functional in Eq.(5) or change the action to a zero space-time disordered (instead of correlated) model (see discussion in the report).
2) Give a proof or a more detailed explanation of Eq.(7) for the detection of unphysical solutions.
3) Correct typos

---

## Round 2 · Referee Report · Anonymous (Referee 1) · 2024-4-9

Strengths

No changes compared to the previous report.

Weaknesses

The two main weaknesses I emphasized in the previous report have been resolved by revision.

Report

The authors make improvements to the manuscript and address the comments of the Referees.

The two main criticisms by the Referees were:

1) only reading the text, one cannot clearly understand what the progress is that is made in the manuscript 2) the criterion for detecting misleading convergence (Eq.9) is not derived in the paper

The authors address both of these criticisms. They add some additional explanations of what is actually new in the present work. In my opinion, the paper presents a reasonable amount of new results, and I expect the paper to be useful to people employing BDMC. The authors also provide a short derivation of Eq.9 in the appendix. I find no issues with the proof, but I feel like one could improve the presentation by adding a few sentences at the end to connect the proof more explicitly with the written statements Eq.9 and Eq.10.

The authors also address the minor criticisms of the Referees. Most importantly, the authors elaborate and give guidelines on how to choose \xi when applying the test for misleading convergence.

In general I think that the manuscript is in good shape, and I recommend the publication of the manuscript in SciPost Physics. In my opinion, the paper satisfies the acceptance criterion "Open a new pathway in an existing or a new research direction, with clear potential for multipronged follow-up work". The paper does not solve the issue of misleading convergence of the self-energy in BDMC methods, but it is equally important to be able to distinguish between correct and incorrect results, and this paper will definitely help with that.

Recommendation

Publish (meets expectations and criteria for this Journal)

---

## Round 2 · Author Response

See the two files
reply.pdf
manuscript_marked_changes.pdf
available at the URL:
https://drive.google.com/drive/folders/1hXSIMyShQfiDlJMMofX7C5Op7KFrJxUl?

---

## Round 2 · List of Changes

Also included in the files provided at the URL given above (in the field "author comments").

---

## Round 3 · Author Response

We thank the Referee for making the following additional suggestion, about the appendix:

``I feel like one could improve the presentation by adding a few sentences at the end to connect the proof more explicitly with the written statements Eq.9 and Eq.10."

We implemented this suggestion by making two minor additions, listed below, and appearing in blue in the updated manuscript.

---

## Round 3 · List of Changes

1) We added the following sentence at the end of the appendix, to reiterate the connection with the criterion Eq.9:

``This concludes the derivation of the equality $G_\infty = G_{\rm exact}$ under the assumption (9)."

2) To connect with the simplified criterion Eq.10:

In the appendix, we prefer to stick to the derivation of the criterion Eq.9 and not to discuss Eq.10 (in accordance with the title of the Appendix and its description in the main text). Instead, we added, in the main text, the following sentence (right after Eq.10):

``Indeed, (9) implies (10), and a situation where (10) would hold while (9) would not hold seems unlikely to occur."

---

## Editorial Decision

accepted_in_target_journal